# Let the Rule Speak: Enhancing In-context Learning Debiasing with Interpretability

## Abstract

In-context learning, which allows large language models to perform diverse tasks with a few demonstrations, is found to have imbalanced per-class prediction accuracy on multi-class text classification. Although notable output correction methods have been developed to tackle the issue and simultaneously improve downstream prediction accuracy, they may fail to answer the core interpretability challenges: why and which certain classes need corrections, and more importantly, a tailored correction for per-sample, per-class's probability. To address such interpretability gaps, we first find that the imbalance arises from certain classes consistently receiving high ICL output probabilities, whereas others receiving lower or mixed ranges, so the former is more frequently chosen, resulting in higher accuracy; more crucially, we find that these ranges have significantly varying degrees of influence on the accuracy bias, highlighting the need for precise, interpretable probability corrections by range. Motivated by this, we propose FuRud, a **Fu**zzy **Ru**le Optimization based **D**ebiasing method, that (1) detects which classes need corrections, and (2) for each correction-needed class, detects its probability ranges and applies asymmetric amplifications or reductions to correct them interpretably. Notably, across seven benchmark datasets, FuRud reduces the pairwise class accuracy bias (COBias) by more than half (56%), while achieving a relative increase of 21% in accuracy, outperforming state-of-the-art debiasing methods. Moreover, FuRud can optimize a downstream task in a few-shot manner, with as few as 10 optimization examples. Furthermore, FuRud can work for prompt formats that lead to highly skewed predictions. For example, FuRud greatly improves ICL outputs which use letter options, with 44% relative accuracy increase and 54% relative COBias reduction.

## 1 Introduction

The classification outputs by in-context learning (ICL) are described as *biased* when they exhibit imbalanced per-class prediction accuracy. Addressing such imbalances while improving overall accuracy is seen as a category of *debiasing*. Concretely, the skewness in the output space can be alleviated by targeted corrections on output logits or probabilities, with or without explicitly modeling the per-class accuracy differences, i.e., COBias (Lin & You, 2024). However, while effective, prior methods could lack straightforward explanations on why and which certain classes need corrections. What's more challenging is to have a tailored per-sample, per-class correction.

A direct cause of COBias is that ICL tends to assign specific ranges of output probabilities to each class. When some classes always receive high probabilities for any input example, others may have lower or mixed probability ranges. The consequence is that latter classes are less frequently predicted than the former, resulting in consistently lower accuracies and calling for probability corrections. In addition, among all examples of a class A, the subset of examples whose in-context learned probability of answer A is relatively low often receive a lower test accuracy, compared to the subset whose class A probability is higher, suggesting that different probability ranges within a class need different corrections.

Taking these overlooked aspects into account, a correction should be tailored for each class and for each sample. To achieve this, a helpful correction should be able to asymmetrically amplify or reduce different ranges of a class's probabilities. In this paper, we address the pressing need for

enhanced understandings in how biased ICL predictions happen, and propose two research questions about a main concern, yet a potential direction, in interpretable ICL output corrections.

**RQ1: What is the interpretability challenge in correcting in-context learned representations?** Given an $N$-class classification dataset, let us denote its $m$-th example's input prompt and label as $(x_m, y_m)$, where $x_m$ consists of a task instruction, few-shot demonstrative examples, and the input example's question. The LLM in-context learns the class probabilities $\boldsymbol{p}_m = (p_{m1}, \ldots, p_{mN})$ (normalized over the $N$ classes), and the prediction $\hat{y}_m$ is $\arg\max_i p_{mi}$. The probabilities $\boldsymbol{p}_m$ may need corrections given the debiasing objective of reducing COBias. Therefore, our task is to correct certain dimensions of $\boldsymbol{p}_m$ towards reducing COBias and improving overall accuracy. The interpretability challenges raised in this process can be specified as (1), detecting which classes need corrections, and (2), for each correction-needed class, applying range-specific amplifications/reductions.

**RQ2: How can we improve interpretability with fuzzy rules?** We leverage membership functions from the field of fuzzy rule based systems for debiasing. For backgrounds, a membership function is a curve that defines a mapping from a crisp input value to a fuzzy value between 0 and 1 (Zadeh, 1965). Based on this, given class probabilities as input attributes, membership functions transform the probabilities to fuzzy values, which could be viewed as corrected probabilities under certain debiasing optimization objectives.

The key intuition here is that a membership function can asymmetrically amplify or reduce different ranges of inputs. Therefore, a fuzzy rule based debiaser for class probability $p_{mi}$ can be written as $f_{A_i}(p_{mi})$, where $A_i$ is a fuzzy set for class $i$, and its membership function $f_{A_i}$ maps the probability to a corrected $p'_{mi} := f_{A_i}(p_{mi})$. Then $\boldsymbol{p}'_m$ consists of corrected per-class probabilities.

Alternatively, the debiaser can be viewed as a **single rule**:

$$\text{If } \underbrace{\text{class 1 is } A_1 \text{ and ... and class } N \text{ is } A_N}_{\text{Antecedent}} \text{ then } \underbrace{\text{predict } \arg\max_j f_{A_j}(p_{mj})}_{\text{Consequent}} \tag{1}$$

Our goal is to optimize the rule, i.e., select fuzzy sets/membership functions for every class in the antecedent, towards mitigating COBias and improving overall prediction accuracy. Specially, we include a *Don't Change* membership function that will keep a class unchanged, suggesting that the LLM in-context learns an accurate probability for the class. When a correction is needed, the membership function detects the probability range that a class's probability belongs to, and updates it with the returned function value. The problem becomes jointly selecting a set of membership functions for each class towards improving multi-objectives based on COBias and accuracy.

To this end, we propose a **Fu**zzy **Ru**le Optimization based **D**ebiasing method, FuRud, which demonstrate via extensive experiments (Section 4) and discussions (Section 5) that it achieves good improvements over accuracy and COBias while providing sample-level interpretability.

In a nutshell, FuRud uses an optimization set of samples for membership function selection. The optimization set's questions are prompted in 1-shot manner, and probabilities are measured across answer classes for each question. These probabilities and ground-truth answers across all questions are aggregated in the multi-objective model, to jointly learn an optimal membership function for each class. At inference, a test example's class probabilities are obtained similarly. Then we apply the learned membership functions to perform tailored corrections at each class's probability in the given test sample. An overview of FuRud is shown in Figure 1, illustrating desired corrections and performance improvements.

To highlight, the membership functions learned by FuRud enable sample-level interpretability. FuRud enables us to know whether the LLM in-context learns an accurate probability for a class within a given sample. This is achieved by learning a correction function (membership function) for each class, towards the multi-objectives of reducing COBias and enhancing accuracy. If the Don't Change function is learned for a class, it means the LLM in-context learns an accurate probability for the class; otherwise, a tailored correction is performed by the membership function. The source code will be released upon paper publication. In summary, our messages are:

- We propose an interpretable fuzzy rule optimization based debiasing method (FuRud), to account for both inter-class surface biases and intra-class range-wise influences.

- We formulate a multi-objective programming model to jointly optimize a set of triangular membership functions for each class. The functions are human-readable, which can asymmetrically correct probabilities of different ranges that are misrepresented.

- Across seven benchmarks, FuRud demonstrates its effectiveness for improved overall accuracy, reduced per-class accuracy imbalance, and enhanced interpretability. For example, it improves ICL accuracy by a relative increase of 21% and reduces COBias by a relative decrease of 56%; it achieves higher accuracy (avg. accuracy reaching 72.0%) and competitive COBias (avg. COBias dropping to 17.8%) over state-of-the-art debiasing methods.

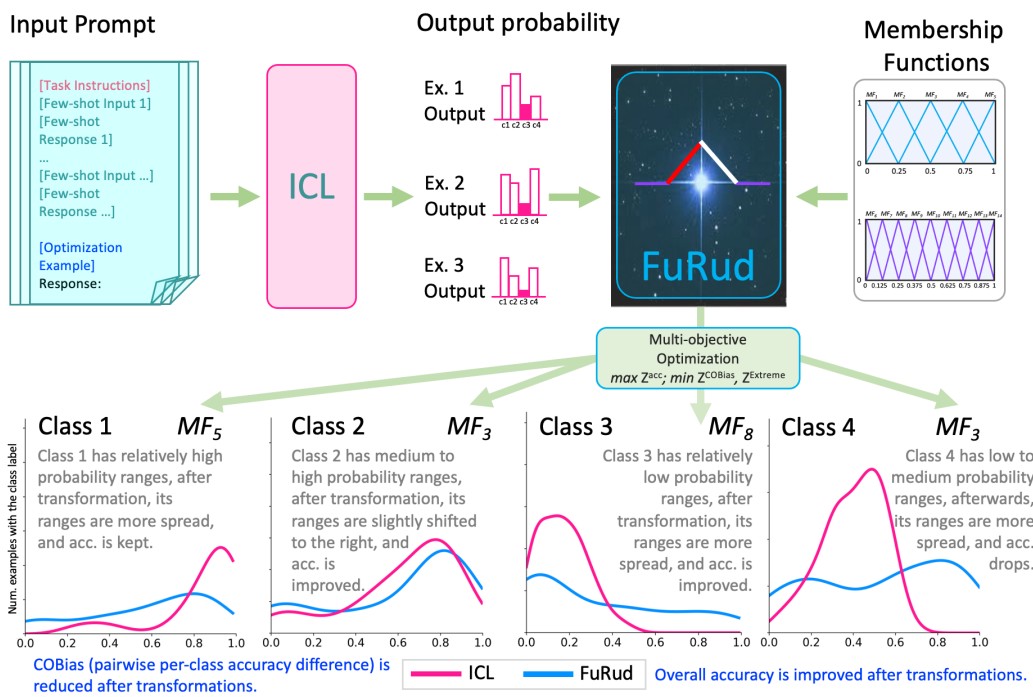

Figure 1: An overview of how FuRud optimizes and transforms each class of a dataset with interpretability; the input to FuRud model is the $N$-dimensional probability vectors of the optimization set of a dataset, and the output is membership functions selected for each class; the selected functions are directly plugged in to test examples at inference. This is for illustration purposes only, actual range changes and improvements vary across datasets.

## 2 RELATED WORK

**Language Model Bias Mitigation.** At the heart of debiasing is detecting biased patterns that arise in a large language model (LLM)'s outputs. Prior work has found various prediction biases in ICL, and address the biased patterns by methods of contextual prompt engineering and output adjustment (Brown et al., 2020; Schick et al., 2021; Zhao et al., 2021). Particularly, on classification tasks, researchers have found that LLMs' outputs are sensitive to ICL formatting, such as prompt templates, demonstrations, and verbalizers (Min et al., 2022; Holtzman et al., 2021; Schick & Schütze, 2021); besides, LLMs tend to output common tokens in the pre-training data (Zhao et al., 2021). These bias factors lead to majority label bias (Zhao et al., 2021), COBias (pairwise class accuracy differences) (Lin & You, 2024), *etc*, causing imbalanced per-class accuracies, and researchers address these biases by making output distribution calibrations (Zhao et al., 2021; Fei et al., 2023; Zhou et al., 2024), or by class probability re-weighting (Lin & You, 2024). For example, Zhao et al. (2021) calibrate the output distribution with content-free/dummy test prompts. Zhou et al. (2024) calibrate the output distribution in a test-time manner, estimating a contextual correction term of each class on a batch of test examples; the proposed Batch Calibration (BC) method outperforms previous calibration methods (Zhao et al., 2021; Fei et al., 2023) on a range of text classification tasks. Lin & You (2024)

re-weights output probabilities by a set of class-specific weight coefficients; the proposed Debiasing as Nonlinear Integer Programming method (DNIP) achieves much lower COBias with higher accuracy than the ICL baseline. Though these debiasing methods effectively adjust ICL outputs, they do not emphasize interpretable bias handling. For example, a calibration method may not explicitly explain why a class needs corrections, or users may not fathom how a re-weighting method performs the exact corrections a class need.

**Fuzzy Rule Techniques for Interpretable Machine Learning.** Interpretable machine learning often needs a human-readable subset of features to generate the target (Jethani et al., 2021; Carvalho et al., 2019). Fuzzy rules are intrinsically interpretable and are widely studied for interpretable machine learning (Vernon et al., 2024; Vilone & Longo, 2020; Ishibuchi & Nojima, 2007). In classical fuzzy rule classification systems, input attributes are assigned to fuzzy sets to generate rules for pattern classification (Ishibuchi et al., 1999; 2005; Nojima & Ishibuchi, 2016; Rudziński, 2016; Gorzałczany & Rudziński, 2017). A fuzzy classification system thus contains multiple human-readable rules, which can be as simple as "1. If attribute Bare Nuclei is *Small* then consequent class *Benign*.2....3. If attribute Uniformity of Cell Size is *not Small* then consequent class *Malignant*." (Gorzałczany & Rudziński, 2017). Here, *Small* and *not Small* are fuzzy sets, with corresponding membership functions. Membership functions provide the core interpretability of the fuzzy systems. In this work, we extend fuzzy membership functions to help with debiasing.

# 3 FuRud: Fuzzy Rule Optimization Based Debiasing

The core idea is to handle the imbalanced per-class accuracy issue with fuzzy membership functions. In the fuzzy rule setting, for $N$ classes, each class selects a fuzzy set $A_i$, or equivalently, a membership function $f_{A_i}$, from a family of $K$ fixed fuzzy sets. We let $F = \{f_1, ..., f_k, ..., f_K\}$ denote the family of membership functions. The membership function selection problem can be solved using combinatorial optimization. To this end, we introduce **FuRud**, a **Fu**zzy **Ru**le Optimization Based **d**ebiasing method. The FuRud optimization is performed on a set of labeled examples, and the selected membership functions are directly applied to transform test-time class probabilities.

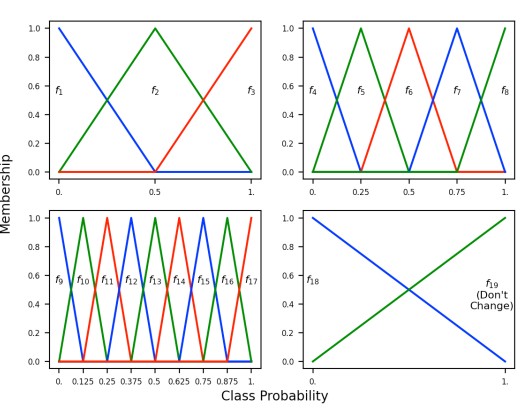

Figure 2: The family of membership functions.

**Membership Functions.** We first introduce the triangular membership functions to select from. Triangular membership functions are popular for fuzzy rule-based classification (Ishibuchi et al., 2005). The main benefits of triangular functions are: the speed of changes is easily controlled by the slope, and the linearity is computationally efficient and easy to understand. Since we do not know an appropriate fuzzy partition for each class in downstream datasets, we simultaneously employ four fuzzy partitions, resulting in membership functions of different granularities.

Figure 2 shows 19 triangular membership functions of four fuzzy partitions, including the *Don't Change* membership function - the identity function (slope=1). Other than *Don't Change*, each membership function represents a sharp or smooth transformation of the input variable. Details of the functions are discussed in Appendix A. The general form of a triangular membership function $f_k(\cdot)$ can be written as:

$$f_k(p_{mi}; a_k, b_k, c_k) = \begin{cases} 0, & \text{if } p_{mi} \leq a_k \\ \dfrac{p_{mi} - a_k}{b_k - a_k}, & a_k \leq p_{mi} \leq b_k \\ \dfrac{c_k - p_{mi}}{c_k - b_k}, & b_k \leq p_{mi} \leq c_k \\ 0, & \text{otherwise} \end{cases} \tag{2}$$

where $a_k, b_k, c_k$ are the left endpoint, the input value where the peak is reached, and the right endpoint of $f_k$. For example, for $f_{11}$, the $a_k, b_k, c_k$ values are 0.125, 0.25, 0.375 respectively.

Then, we compute the updated probability $p'_{mi}$ by:

$$p'_{mi} = \begin{cases} p_{mi}, & \text{if } \sum_{i=1}^{N} p'_{mi} = 0 \\ \sum_k f_k(p_{mi}) \mathbb{1}(\kappa_i = k), & \text{otherwise} \end{cases} \tag{3}$$

where $\kappa_i$ is the integer selection variable for class $i$. $\mathbb{1}(\cdot)$ evaluates to 1 if the condition inside is satisfied, otherwise 0. Furthermore, in case $p'_{mi} = 0$ for all classes, we reset each to be its original probability in $\boldsymbol{p}_m$. Therefore, $\hat{y}_m = \arg\max_i p'_{mi}$.

**Multi-Objective Programming and Energy Function.** Let $\boldsymbol{\kappa} = (\kappa_1, \ldots, \kappa_N)$ be the integer selection variables for classes $1, ..., N$, where $\kappa_i$ is chosen from the given set of membership functions, and $\kappa_i = k$ means $f_k$ is chosen. Our goal is to learn $\boldsymbol{\kappa}$ that improve ICL classifications under two main evaluation metrics, accuracy and COBias (Lin & You, 2024). To this end, we adopt multi-objective programming for simultaneous better accuracy and lower COBias.

The first objective is to improve overall accuracy:

$$\max Z^{\text{Acc}} = \frac{1}{|S^{\text{Opt}}|} \sum_{m \in S^{\text{Opt}}} \mathbb{1}\{\hat{y}_m = y_m\} \tag{4}$$

where $S^{\text{Opt}}$ is the indices of examples used for optimization.

Furthermore, we balance the class accuracy difference by explicitly modeling COBias, which accounts for an overall difference between pairwise per-class accuracies. Minimizing COBias helps address low-accuracy classes from ICL outputs. Therefore, the second objective is:

$$\min Z^{\text{COBias}} = \frac{1}{{}_N C_2} \sum_{i=1}^{N-1} \sum_{j=i+1}^{N} \left| \text{Acc}_i - \text{Acc}_j \right| \tag{5}$$

where ${}_N C_2 = N(N-1)/2$, $\text{Acc}_i$ is the accuracy score for optimization examples in class $i$.

To further handle extreme cases of low class accuracies, we penalize classes that fail to reach an accuracy threshold, and minimize the loss between the threshold and per-class accuracy (cut off at 0). The third objective is:

$$\min Z^{\text{Extreme}} = \sum_{i=1}^{N} \max\{0, \lambda - \text{Acc}_i\} \tag{6}$$

where $\lambda$ is a fixed threshold value.

The above objective functions are a mix of minimization and maximization, and the resulted multi-objective programming model requires integer variables. Each of them alone corresponds to an integer programming problem, which is NP-complete (Garey & Johnson, 1979). Classic solutions for integer programming use operational research techniques, such as Branch-and-Bound, often used for linear integer programming problems. It could be difficult for such methods to handle nonlinear integer programming models which contain non-differentiable functions. Consequently, a series of metaheuristic algorithms have emerged, such as Simulated Annealing (SA), and each metaheuristic has their own strengths and limitations. We use one of the metaheuristics, SA, to tackle the proposed mathematical model. The SA implementation follows (Lin & You, 2024). Since it is difficult to solve each one as an individual optimization problem and force an optimal solution, our strategy is instead to compute a weighted sum of $1 - Z^{\text{Acc}}, Z^{\text{COBias}}, Z^{\text{Extreme}}$ as a single energy function $E$ to be optimized using SA. Hence, the multi-objectives are combined into a total minimization objective:

$$\min_{\kappa} E(\kappa; \lambda, \boldsymbol{p}') \tag{7}$$

where $E(\kappa; \lambda, \boldsymbol{p}') = \omega + \sum_{h \in S^{\text{Obj}}} \gamma^h Z^h$, $S^{\text{Obj}}$ is the names of the penalty functions corresponding to the individual objectives, and $\omega, \gamma^h$s are penalty parameters. Therefore, the SA algorithm optimizes on $E$ to obtain an optimal set of membership functions.

In summary, the class corrections aim at reducing COBias and improving accuracy. Each equation from 4 to 6 exactly targets one of these two goals. In detail, Eq. 4 targets maximizing overall accuracy, Eq. 6 targets minimizing COBias, and Eq. 6 targets maximizing per-class accuracy, which enforces it to meet a threshold; Eq. 7 combines the three objectives as a multi-objective function. Details on how Eq. 7 is optimized are described in experimental setups (Section 4.1).

## 4 EXPERIMENTS

### 4.1 EXPERIMENTAL SETUPS

**Evaluation Tasks and Evaluation Metrics.** The proposed method is evaluated on a diverse range of text classification datasets, including AGNews (Zhang et al., 2015), a 4-class news topic classification; DBpedia (Auer et al., 2007), a 14-class ontology classification dataset derived from Wikipedia; SST-5 (Socher et al., 2013), a 5-class sentiment classification dataset; TREC (Voorhees & Tice, 2000; Li & Roth, 2002), a 6-class question classification dataset; RTE (Dagan et al., 2006), a binary entailment recognition dataset; and two biomedical domain-specific datasets, including DDI (Segura-Bedmar et al., 2013), a 5-class drug-drug interaction relation extraction dataset; PubMedQA (Jin et al., 2019), a 3-class biomedical question answering dataset. Each evaluation dataset is split into optimization/development/test sets. We follow (Lin & You, 2024) to preprocess the datasets. Evaluation metrics are accuracy and COBias.

**FuRud Setups.** The 19 triangular membership functions in Figure 2 form the base of selections for FuRud. To obtain the per-class probabilities from ICL, we prompt Llama-2-13B (13B parameters) in 1-shot manner. The output softmax probabilities normalized over all classes are used as attributes. The energy function we used in the experiments is a special form of Equation 7 with $\omega = 1, \gamma^{\text{Acc}} = -1, \gamma^{\text{COBias}} = \alpha, \gamma^{\text{Extreme}} = \beta$. In other words, the final multi-objective optimization function is $min_\kappa Z = 1 - Z^{\text{Acc}} + \alpha Z^{\text{COBias}} + \beta Z^{\text{Extreme}}$, where we learn $\kappa_i$ for class $i = 1, \ldots, N$ on an optimization set of samples, which is the full or a subset of training set. Each $\kappa_i$ is selected from the given set of membership functions, and $\kappa_i = k$ means membership function $f_k$ is selected. At inference time, for a test sample, let $p = (p_1, \ldots, p_i, \ldots, p_N)$ be its in-context learned output class probabilities, then these probabilities are transformed by their learned membership functions, according to Eq. 3. The corrected prediction is $\hat{y} = \arg\max_i f_{\kappa_i}(p_i)$.

The above model $Z$ is optimized using the SA metaheuristic. The core step of SA is to sample a new solution $\kappa = (\kappa_1, \ldots, \kappa_N)$, e.g., $(16, \ldots, 8)$, and evaluate it on $Z$. If $Z$ is smaller, the algorithm accepts the new solution; otherwise, it accepts the new solution with an acceptance probability $exp(-\Delta Z/T)$, where $T$ is the temperature at the step. The values of $\alpha, \beta$ are tuned on the development set. Since we do not know an estimate for the expected threshold value $\lambda$ in downstream tasks, we set it to 0.5 for simplicity. Prompting is done on a 80G A100 GPU. The simulated annealing algorithm executes on an AMD EPYC 7742 CPU with execution time in minutes.

We compare FuRud with the ICL baseline and two state-of-the-art ICL debiasing methods, including DNIP (Lin & You, 2024) and BC (Zhou et al., 2024). For fair comparisons, for each dataset, we prompt with three different 1-shot demonstrations and obtain three sets of initial probabilities. The demonstration is randomly sampled from optimization examples. The average test accuracy and COBias over the three runs are reported.

### 4.2 MAIN RESULTS

| Method | Acc. ↑ | | | | COBias ↓ | | | |
|---|---|---|---|---|---|---|---|---|
| | ICL | BC | DNIP | FuRud | ICL | BC | DNIP | FuRud |
| AGNews | $79.9_{7.0}$ | $82.5_{5.0}$ | $87.9_{0.7}$ | $85.7_{3.4}$ | $28.3_{16.1}$ | $23.1_{12.1}$ | $6.3_{0.6}$ | $6.9_{1.6}$ |
| DBpedia | $88.6_{1.7}$ | $89.1_{1.5}$ | $93.4_{0.6}$ | $92.2_{0.4}$ | $16.2_{3.7}$ | $15.4_{3.3}$ | $7.7_{0.6}$ | $9.2_{0.6}$ |
| SST-5 | $44.9_{4.3}$ | $47.6_{2.3}$ | $48.3_{1.9}$ | $48.8_{3.8}$ | $53.1_{5.0}$ | $49.8_{10.7}$ | $18.7_{10.1}$ | $22.2_{8.4}$ |
| TREC | $68.5_{10.8}$ | $72.9_{4.4}$ | $77.1_{2.0}$ | $77.3_{3.9}$ | $35.9_{6.5}$ | $31.9_{5.1}$ | $14.2_{1.3}$ | $18.5_{1.4}$ |
| RTE | $71.5_{2.2}$ | $76.1_{0.6}$ | $74.3_{0.8}$ | $74.5_{1.8}$ | $43.7_{7.0}$ | $16.4_{1.9}$ | $4.3_{3.3}$ | $7.1_{5.0}$ |
| DDI | $7.2_{0.9}$ | $14.4_{2.5}$ | $40.4_{6.0}$ | $69.3_{6.3}$ | $45.6_{5.9}$ | $32.6_{7.6}$ | $7.5_{3.2}$ | $36.8_{4.6}$ |
| PubMedaQA | $55.1_{2.9}$ | $55.5_{1.3}$ | $63.1_{14.0}$ | $55.9_{5.4}$ | $61.2_{1.9}$ | $26.2_{3.2}$ | $41.1_{29.6}$ | $24.0_{8.4}$ |
| Avg. | 59.4 | 62.6 | 69.2 | **72.0** | 40.5 | 27.9 | **14.3** | 17.8 |

Table 1: Test accuracy and COBias (%); average scores over three runs are reported. FuRud outperforms previous methods in accuracy, and is on par with DNIP in COBias.

Table 1 shows the test accuracy and COBias of ICL, BC, DNIP, and FuRuD. Comparing FuRud to the ICL baseline, the average relative accuracy increase is 21%, and the average relative CO-Bias reduction is 56%. The average test accuracy of FuRud over seven benchmarks is 72%, which outperforms the accuracy of BC and DNIP; the average test COBias of FuRud is 17.8%, which is comparable to DNIP with obtains the lowest COBias (14.3% ) among the methods compared. It is noted that FuRud uses the full optimization set to make a fair comparison to DNIP. However, FuRud can also work in a few-shot optimization manner, as discussed in Section 5.2. On top of that, FuRud provides enhanced interpretability, as visualized in the following section.

## 4.3 INTERPRETABILITY ANALYSIS

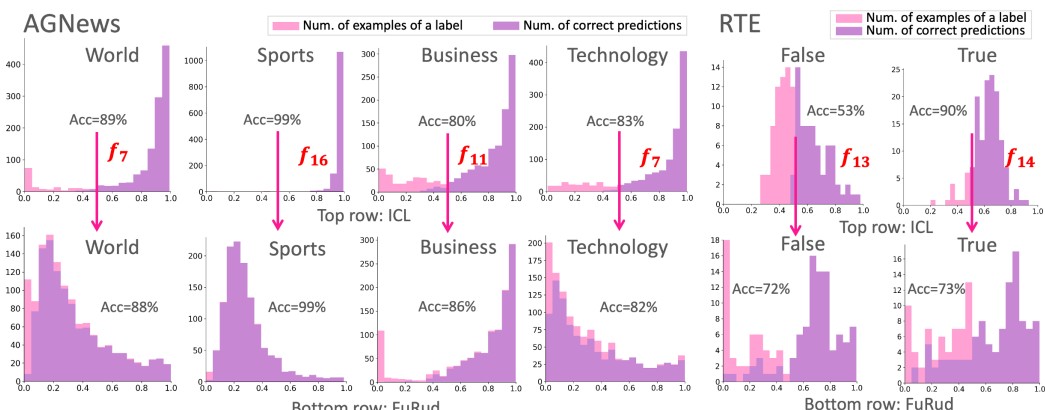

Figure 3: Class probability changes before and after applying FuRud. There was a stark accuracy difference of 37% for RTE's *True* and *False* before FuRud, manifesting the model (ICL)'s tendency to assign higher probabilities to *True*. FuRud addresses this accuracy bias by amplifying the medium range of *False* and simultaneously reducing the relatively high range of *True*.

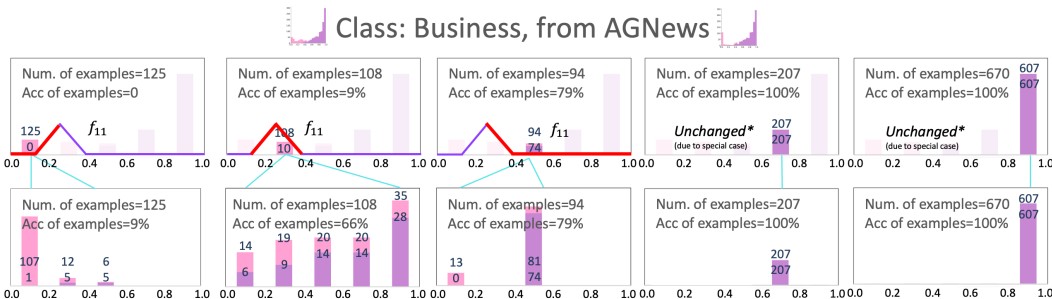

Top row: number of examples in Business and accuracy (proportion of purple) of five different probability ranges, by ICL (scaled relatively).
Bottom row: new ranges and improved accuracy for the examples in each previous range, by FuRud (scaled relatively), suggesting that examples after fuzzy transformations have more accurate output probabilities for class Business.

Figure 4: Zooming in on transformations applied to class *Business* from AGNews, whose accuracy increases from 80% (ICL) to 86%. The special case returns the original class probability of an example when transformed probabilities sum to 0 (Eq. 3).

We visualize the class-wise probability changes before and after applying FuRud in Figure 3. AG-News and RTE are taken as examples (other datasets' results are similar). The run with seed 1 out of all three runs is used for illustrating the membership functions. For both AGNews and RTE, around half of the classes have an increased/kept accuracy. More importantly, on both datasets, the worst-performing class by ICL significantly improves. In details, the relatively low to medium probability ranges of the worst-performing class gets amplified, whereas the relatively high probability ranges of other classes gets slightly reduced. This shows FuRud's effective amplifications or reductions in the most correction-needed probability ranges of a class.

To further see this, Figure 4 illustrates the detailed transformation of different probability ranges of class *Business* of AGNews. For the 1,204 test examples with label *Business*, we divide their ICL output probabilities at the position of class *Business* into 5 different ranges, from $[0.0, 0.2]$ to $[0.8, 1.0]$. The top row shows that examples in the first two ranges, or $[0.0, 0.4]$, have relatively low accuracies (0 and 9%). These probabilities need corrections most, which are effectively transformed by the membership function $f_{11}$, selected by FuRud for class *Business*. The red color highlights activated parts for the transformations, resulting in new probability ranges of the examples and improved accuracies (9% and 66%). This further demonstrates the improved interpretability and higher accuracy obtained by FuRud, especially for a less performing class.

## 5 DISCUSSION

### 5.1 FURUD GREATLY IMPROVES HIGHLY SKEWED LETTER BASED ICL OUTPUTS, BY 44% RELATIVE ACCURACY INCREASE AND 54% RELATIVE COBIAS REDUCTION

In this section, we show the effectiveness of FuRud under a different set of prompt output choices - the letter options, which could lead to more serious shallow matching issue than label token options. When letter options are used in a prompt, a model is expected to output a single letter choice of "A", "B", *etc.* mapping to a class label. Output choices significantly contribute to prompt sensitivity. In fact, LLMs have been shown to have a tendency to select a certain letter option regardless of the content, where for instance a model could over-predict the letter

| Method | Acc. | COBias |
|--------|------|--------|
| ICL (letter) | $36.9_{13.6}$ | $47.2_{15.6}$ |
| FuRud (letter) | $53.1_{10.5}$ | $21.6_{8.2}$ |

Table 2: Test Scores (%) of FuRud on Letter Based ICL Outputs, averaged over the seven datasets.

"A" (Bentham et al., 2024), suggesting moderate to high COBias. This surface pattern matching issue of letter options is also obvious on the datasets we evaluated, which could even lead to over 90% accuracy in the biased class and much lower accuracy in some other classes. For example, on AGNews, the model is biased to predict "B" (class label: *Sports*), leading to an average of 99% accuracy in *Sports* and 12% accuracy in *Business* over three runs.

We apply FuRud to the highly distorted letter based ICL outputs. Table 2 shows the test accuracy and COBias for ICL and FuRud, averaged over seven benchmark datasets, where FuRud improves accuracy by an relative 44% and achieves a significant COBias reduction of a relative 54% over ICL. Besides the tabled results, on the aforementioned AGNews dataset, overall test accuracy improves to 66% from 45%, and COBias reduces to 10% from 54%. The per-class accuracy changes from ICL to FuRud are: *World*, 40% → 69%; *Sports*, 99% → 70%; *Business*, 12% → 66%; *Technology*, 27% → 59%. These results demonstrate the effectiveness of FuRud on debiasing highly skewed ICL outputs, suggesting that FuRud can debias no matter how poor or perfect the input prompt is.

### 5.2 FEW-SHOT OPTIMIZATION

FuRud can optimize a downstream task with as few as 10 examples. Figure 5 shows test accuracy and COBias of FuRud (in mint green color) when used in a few-shot optimization manner, starting with 10 few-shot examples and growing to 100 and 500 examples. TREC and SST-5 are shown to illustrate that FuRud can achieve an average of 9% accuracy improvements with 18% COBias reduction over the ICL baseline at 10 few-shot optimization examples. At 10 examples, FuRud obtains a 11% and 6% relative increase in accuracy over the ICL baseline

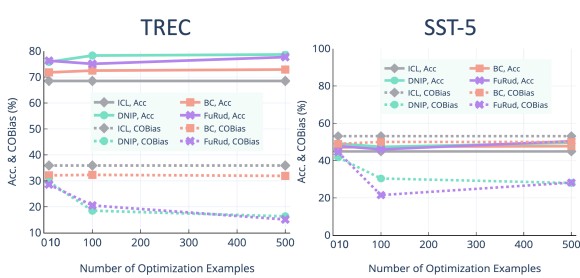

Figure 5: Few-shot optimization.

on TREC and SST-5 respectively, at the same time, it reduces COBias by a relative 20% and 16% on each dataset. The accruacy and COBias performances gradually improve as the number of examples increases to 500. Compared to existing methods, FuRud outperforms BC in few-shot scenarios,

and performs better than (TREC) or on par (SST-5) with DNIP while being interpretable. Similar findings apply to the other five datasets, as shown in Appendix B.

## 5.3 EFFECT OF MEMBERSHIP FUNCTION GRANULARITIES

We experiment with different combinations of the four fuzzy partitions in Figure 2, in addition to the main results using all partitions. The partitions are characterized by different rates of change, i.e., different absolute values of slopes of the rising/falling edges. A larger slope indicates more granularities. The slopes for the top left, top right, bottom left, and bottom right partitions are $\pm 1, \pm 2, \pm 4, \pm 8$ respectively. Specifically, the bottom right partition has the *Don't Change* function $y = x$ and its symmetric function $y = 1 - x$, which will be referred to as the DC partition. Since the *Don't Change* function plays a vital role in keeping some classes unchanged, we experiment with five combinations, including DC , and DC with each partition of slope $\pm 2, \pm 4, \pm 8$. The accuracy and COBias scores of five combinations are shown in Figure

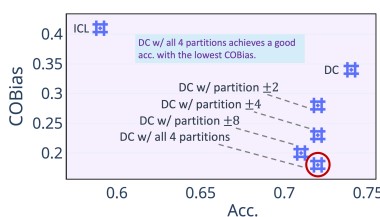

Figure 6: Accuracy-COBias trade-off with 5 combinations of fuzzy partitions.

6. The average score of seven datasets are reported, and for each dataset, the average accuracy and COBias over three runs is taken. COBias reduces with higher granularities and accuracy slightly decreases. DC can reach 74% accuracy, being 15% higher than ICL accuracy, but the improvement is mainly from DDI, suggesting that DC alone is not enough to transform the biased probabilities. The optimal accuracy and COBias is achieved with mixed partitions.

In addition, the *Don't Change* fuction is essentially needed in debiasing. We perform an ablation analysis with the partition $\pm 8$ only, and find that, while achieving similar accuracies, its COBias is 6% higher than using DC with partition $\pm 8$. Moreover, for example, 4 out of 14 classes on DBpedia are optimized with *Don't Change*, suggesting that keeping certain classes unchanged is necessary for jointly optimizing overall accuracy and COBias. This demonstrates that a dedicated *Don't Change* function is needed in the multi-objective optimization.

In summary, higher membership function granularities are good for COBias reduction. However, although it is tempting to include as many membership functions as possible to reduce COBias, there is the accuracy-COBias tradeoff. Too many membership functions may not further boost accuracy and could induce more computational costs.

## 5.4 MORE DISCUSSIONS

**FuRud's Performances on More LLMs.** For more LLMs of varied sizes and families, FuRud consistently improves both overall accuracy and COBias, showcased by the additional experimental results on Llama-2-7B and GPT-2-XL in Appendix C.

**FuRud's Performances under More ICL Demonstration Selection Strategies.** To further see how demonstrations in the prompt affect performances, we additionally prompt Llama-2-13B with an additional demonstration selection strategy, k-shot prompting, where k is the number of classes; a demonstrative example from each class is randomly selected from the optimization set, and these examples are cascaded as a demonstrative example. FuRud significantly improves accuracy and COBias in this setting, as detailed in Appendix D.

**Computational Costs.** As for computational costs, the computational time of FuRud optimization is in the scale of minutes, from several minutes to around 30 minutes, depending on the dataset (e.g., number of classes, optimization set sizes, etc). For DNIP, the computational time is similarly in the scale of minutes. For the calibration method Batch Calibration (BC), it applies an analytical calculation on all samples' ICL probabilities, introducing small computational overhead.

**Interpretability compared: DNIP and FuRud.** The DNIP method shows good debiasing performances, but it applies indiscriminate reduction (or relative amplification) to the probabilities, making it difficult to capture the varying degrees of influence of different probability ranges to the accuracy bias, potentially limiting its interpretability. The use of fuzzy membership functions overcomes this issue, and this is a main innovation of our paper.

**Can we use the traditional fuzzy rule based systems for debiasing?** That would require maintaining multiple candidate rules like "$R_q$: If the probability of class 1 is $A_{q1}$ and ... and the probability of class $N$ is $A_{qN}$," then predict $Y_q$," where $Y_q$ is the consequent/target class. Training such rules is computationally expensive, and inference time for a winning rule grows with the number of candidates. Additionally, calculating the product of membership values could cause issues such as overflow, and achieving high accuracy might demand an overwhelming number of rules, making the system inefficient. In contrast, FuRud eliminates the need for learning multiple rules, as its transformations could implicitly capture many rules found in traditional fuzzy classification systems.

**We have a different motivation from traditional post-hoc corrections.** Some may argue that ensuring equitable accuracies across all classes is a well-studied problem in standard machine learning classifiers. It is worth emphasizing that the per-class prediction accuracy imbalance should be treated within their particular context. The accuracy bias in ICL outputs stems from completely different causes than the unequal class accuracies observed in potentially overfitted traditional classifiers, where the former is rooted in prompts and the LLMs, and the latter arises from class imbalance of supervised training data. That's why our method is particularly applied to ICL's output token class probabilities, pinpointing specific patterns and applying precise, targeted corrections.

## 6    CONCLUSION AND FUTURE WORK

In this work, we present a fuzzy rule optimization based debiasing method to enhance ICL output class representations with interpretability. FuRud learns a per-class correction function, i.e., a membership function, which decides if and how a class's probability needs correction for each sample. If correction is needed, the corrected class probability will be tailored by the membership function, which is a main innovation of this paper. On a diverse set of text classification benchmarks, FuRud greatly improves the average test accuracy and test COBias over ICL, by a relative increase of 21% and a relative reduction of 56%, outperforming state-of-the-art methods. Moreover, FuRud can work for prompt formats that may lead to highly skewed predictions, e.g., letter options. Furthermore, FuRud can optimize a downstream task with as few as 10 optimization examples.

In the future, more versatile rules can be explored, and we may also examine the tradeoff between the accuracy and rule complexity. Simpler rules are easier to understand, but the transformations may fail to catch the intricate interactions between class predictions. More complex rules may have better modeling capabilities, but they are harder to read. In addition, this work focuses on evaluating text classification, and we will extend interpretable ICL debiasing to more language tasks, modalities, and model architectures.

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

## A DETAILS ON MEMBERSHIP FUNCTIONS

Table 3 lists the details about the membership functions used in this work.

| Function | Parameters | Name | Short Form | Meaning |
|---|---|---|---|---|
| $f_1$ | 0, 0, 0.5 | Low-2 | L-2 | Low-range transformation, smooth change with slope $-2$, peak at 0 |
| $f_2$ | 0, 0.5, 1 | Medium-2 | M-2 | Medium-range transformation, smooth change with slope $\pm 2$, peak at 0.5 |
| $f_3$ | 0.5, 1, 1 | High-2 | H-2 | High-range transformation, smooth change with slope 2, peak at 1 |
| $f_4$ | 0, 0, 0.25 | Low-4 | L-4 | Low-range transformation, sharp change with slope $-4$, peak at 0 |
| $f_5$ | 0, 0.25, 0.5 | Medium Low-4 | ML-4 | Low-to-medium-range transformation, sharp change with slope $\pm 4$, peak at 0.25 |
| $f_6$ | 0.25, 0.5, 0.75 | Medium-4 | M-4 | Medium-range transformation, sharp change with slope $\pm 4$, peak at 0.5 |
| $f_7$ | 0.5, 0.75, 1 | Medium High-4 | MH-4 | Medium-to-high-range transformation, sharp change with slope $\pm 4$, peak at 0.75 |
| $f_8$ | 0.75, 1, 1 | High-4 | H-4 | High-range transformation, sharp change with slope 4, peak at 1 |
| $f_9$ | 0, 0, 0.125 | Very Very Low-8 | VVL-8 | Very-very-low-range transformation, very sharp change with slope $-8$, peak at 0 |
| $f_{10}$ | 0, 0.125, 0.25 | Very Low-8 | VL-8 | Very-low-range transformation, very sharp change with slope $\pm 8$, peak at 0.125 |
| $f_{11}$ | 0.125, 0.25, 0.375 | Low-8 | L-8 | Low-range transformation, very sharp change with slope $\pm 8$, peak at 0.25 |
| $f_{12}$ | 0.25, 0.375, 0.5 | Medium Low-8 | ML-8 | Low-to-medium-range transformation, very sharp change with slope $\pm 8$, peak at 0.375 |
| $f_{13}$ | 0.375, 0.5, 0.625 | Medium-8 | M-8 | Medium-range transformation, very sharp change with slope $\pm 8$, peak at 0.5 |
| $f_{14}$ | 0.5, 0.625, 0.75 | Medium High-8 | MH-8 | Medium-to-high-range transformation, very sharp change with slope $\pm 8$, peak at 0.625 |
| $f_{15}$ | 0.625, 0.75, 0.875 | High-8 | H-8 | High-range transformation, very sharp change with slope $\pm 8$, peak at 0.75 |
| $f_{16}$ | 0.75, 0.875, 1 | Very High-8 | VH-8 | Very-high-range transformation, very sharp change with slope $\pm 8$, peak at 0.875 |
| $f_{17}$ | 0.875, 1, 1 | Very Very High-8 | VVH-8 | Very-very-high-range transformation, very sharp change with slope 8, peak at 1 |
| $f_{18}$ | 0, 0, 1 | Full-1 | F-1 | Full-range transformation, very smooth change with slope $-1$, peak at 0 |
| $f_{19}$ | 0, 1, 1 | Don't Change | Don't Change | Identity function |

Table 3: Names, parameters $(a, b, c)$, short forms, and meanings for membership functions.

## B ADDITIONAL FEW-SHOT OPTIMIZATION RESULTS

Figure 7 shows additional few-shot optimization results. In a few-shot optimization manner, FuRud achieves better or comparable results than DNIP, and better results than BC and the ICL baseline, while providing enhanced interpretability.

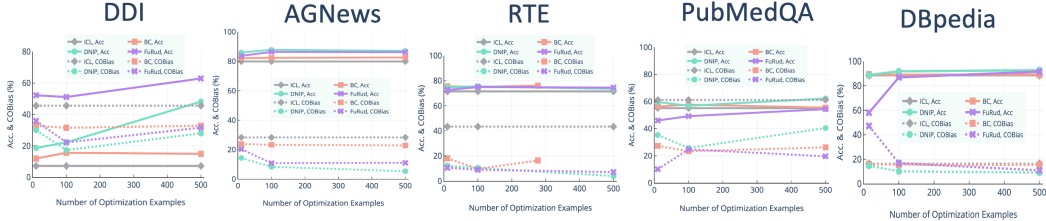

Figure 7: Additional few-shot optimization results.

| Model | Metric | AGNews | DBpedia | SST-5 | TREC | RTE | DDI | PubMedQA | Avg. |
|-------|--------|--------|---------|-------|------|-----|-----|----------|------|
| | | | | Llama-2-7B | | | | | |
| ICL | Acc | $86.4_{2.5}$ | $88.9_{2.0}$ | $42.1_{11.1}$ | $66.7_{6.6}$ | $66.3_{4.3}$ | $6.7_{0.4}$ | $40.3_{6.7}$ | $56.8$ |
| | COBias | $14.0_{6.5}$ | $13.5_{2.1}$ | $55.6_{1.5}$ | $33.2_{10.0}$ | $61.6_{10.5}$ | $41.4_{1.7}$ | $40.9_{16.1}$ | $37.2$ |
| FuRud | Acc | $\mathbf{88.5_{0.5}}$ | $\mathbf{91.5_{0.5}}$ | $\mathbf{49.5_{0.7}}$ | $\mathbf{73.1_{3.9}}$ | $\mathbf{72.7_{1.0}}$ | $\mathbf{54.4_{6.4}}$ | $\mathbf{55.7_{7.6}}$ | $\mathbf{69.3}$ |
| | COBias | $\mathbf{7.4_{2.5}}$ | $\mathbf{8.4_{0.6}}$ | $\mathbf{24.0_{1.2}}$ | $\mathbf{14.1_{1.9}}$ | $\mathbf{4.2_{2.7}}$ | $\mathbf{16.9_{5.0}}$ | $\mathbf{21.8_{16.6}}$ | $\mathbf{13.8}$ |
| | | | | GPT2-XL | | | | | |
| ICL | Acc | $52.1_{5.4}$ | $31.8_{9.9}$ | $34.9_{13.7}$ | $27.4_{10.5}$ | $55.4_{1.9}$ | $14.5_{4.4}$ | $55.2_{0.0}$ | $38.8$ |
| | COBias | $35.5_{11.5}$ | $40.0_{3.6}$ | $48.7_{5.4}$ | $45.6_{8.7}$ | $82.4_{24.5}$ | $40.7_{5.9}$ | $59.4_{12.6}$ | $50.3$ |
| FuRud | Acc | $\mathbf{69.0_{0.5}}$ | $\mathbf{67.7_{11.8}}$ | $\mathbf{43.4_{3.1}}$ | $\mathbf{41.7_{2.7}}$ | $\mathbf{51.2_{3.7}}$ | $\mathbf{53.2_{17.0}}$ | $\mathbf{48.4_{0.3}}$ | $\mathbf{53.5}$ |
| | COBias | $\mathbf{7.4_{2.9}}$ | $\mathbf{23.0_{6.5}}$ | $\mathbf{25.4_{1.4}}$ | $\mathbf{30.2_{7.0}}$ | $\mathbf{8.9_{3.6}}$ | $\mathbf{23.1_{6.5}}$ | $\mathbf{17.6_{4.6}}$ | $\mathbf{19.4}$ |

Table 4: Test accuracy and COBias Comparisons on more LLMs.

## C  FURUD'S PERFORMANCES ON MORE LLMS

We ran experiments of FuRud on two additional models, Llama-2-7B and GPT2-XL. Results are shown in Table 4. For example, on Llama-2-7B, FuRud improves accuracy by a relative 22%, and reduces COBias by a relative 63% over ICL baselines, demonstrating that FuRud gains consistent performance improvements on various models. Indeed, our current evaluations are focused on relatively small LLMs, but our approach can also work for larger models, as long as class probabilities are available and the imbalanced per-class accuracy issue exists.

## D  FURUD'S PERFORMANCES UNDER MORE ICL DEMONSTRATION SELECTION STRATEGIES

| Demonstration Selection | Metric | AGNews | DBpedia | SST-5 | TREC | RTE | DDI | PubMedQA | Avg. |
|-------------------------|--------|--------|---------|-------|------|-----|-----|----------|------|
| k-shot, ICL | Acc | $83.5_{1.5}$ | $95.2_{1.2}$ | $50.3_{2.3}$ | $67.0_{12.7}$ | $75.0_{0.8}$ | $9.7_{1.0}$ | $52.3_{5.3}$ | $61.9$ |
| | COBias | $14.9_{5.1}$ | $7.0_{2.2}$ | $36.3_{7.2}$ | $38.2_{5.1}$ | $22.5_{13.2}$ | $39.7_{3.5}$ | $20.9_{4.2}$ | $25.6$ |
| k-shot, FuRud | Acc | $\mathbf{88.1_{0.6}}$ | $\mathbf{96.6_{0.4}}$ | $\mathbf{54.3_{1.3}}$ | $\mathbf{77.9_{6.0}}$ | $\mathbf{75.9_{4.6}}$ | $\mathbf{62.3_{2.1}}$ | $\mathbf{59.2_{5.9}}$ | $\mathbf{73.5}$ |
| | COBias | $\mathbf{7.7_{2.5}}$ | $\mathbf{4.4_{0.7}}$ | $\mathbf{13.8_{4.1}}$ | $\mathbf{11.6_{3.3}}$ | $\mathbf{5.0_{1.4}}$ | $\mathbf{27.0_{2.2}}$ | $\mathbf{21.3_{8.7}}$ | $\mathbf{13.0}$ |

Table 5: Test accuracy and COBias under the k-shot demonstration selection strategy.

We additionally prompt Llama-2-13B with the following demonstration selection strategy: k-shot prompting, where k is the number of classes. A demonstrative example from each class is randomly selected from the optimization set and represented in the prompt. FuRud significantly improves accuracy and COBias over ICL baselines, as shown in Table 5.

Compared to the 1-shot strategy (Table 1), the k-shot strategy provides a different starting point for FuRud. For example, the average ICL accuracy by k-shot (61.9%) is slightly larger than that obtained by 1-shot (59.4%), and average COBias (25.6%) is smaller than 1-shot (40.5%). FuRud boosts average accuracy to 73.5% and reduces COBias to 13.0%. In conclusion, different example selection strategies provide different starting points for FuRud to optimize, on which FuRud consistently improve.

