# OpenReview forum: "Let the Rule Speak: Enhancing In-context Learning Debiasing with Interpretability"
_ICLR.cc/2025/Conference — Submitted to ICLR 2025_

### Official Review · Reviewer_DNty · 2024-10-30

**Soundness:** 2
**Presentation:** 2
**Contribution:** 2
**Rating:** 3
**Confidence:** 3

**Summary:**

This work aims to promote ICL's performance. In particular, the authors focus on the imbalanced prediction issue. To this end, the authors propose a fuzzy rule optimization-based debiasing method. Some experiments are conducted to evaluate the proposed methods.

**Strengths:**

1. The studied topic is promising, as the imbalanced prediction issue of ICL poses significant challenges to the community.

2. Introducing fuzzy rules is novel and exciting.

**Weaknesses:**

1. I fail to locate the definition of "per-class accuracy bias" throughout the paper, which makes the work difficult to follow. Specifically, what is per-class accuracy bias? What is the difference between the standard scenario and the mentioned per-class accuracy bias?

2. I did not figure out how to optimize Eqs. 4-7 since no clear explanation is provided. The authors did not give a clear picture of the proposed method. Consequently, reproducing this work will be challenging.

3. The performance gain is limited, especially when comparing FuRud to DNIP. For instance, DNIP achieves 6.6\% performance gain over BC, while FuRud is 6% more accurate than DNIP. Similar cases can be found in the COBias metric.

4. The motivation of the proposed method is confusing. I cannot figure out why the proposed method works and why methods are designed using such an approach. In particular, the authors should clarify why Eqs. 4-7 can address the challenge of class correction. Since there is space for one page to allow authors to add necessary explanations, authors may consider adding detailed clarifications.

**Questions:**

Which modules in the experiments or the proposed method are related to Eqs 4-7?

What is the explicit connection between the proposed method and the mentioned interpretability?

---

> ### Author Response · Authors · 2024-11-21
> **Response to Reviewer DNty**
>
> Dear Reviewer DNty,
>
> Thank you for your time, high recognition of our innovation, and constructive suggestions. We are encouraged that you find the topic is promising and introducing fuzzy rules is novel and exciting.
>
> In the following, we would like to address your concerns.
>
> ---
> **Q1**: I fail to locate the definition of "per-class accuracy bias" throughout the paper, which makes the work difficult to follow. Specifically, what is per-class accuracy bias?
>
> **R1**: Thanks for the feedback. Indeed, we wrote "per-class accuracy bias" when we meant per-class accuracy differences, i.e., the imbalanced class accuracy issue. We will revise it to “the imbalanced class accuracy issue”.
>
>
> ---
> **Q2**: I did not figure out how to optimize Eqs. 4-7 since no clear explanation is provided. Consequently, reproducing this work will be challenging. Which modules in the experiments or the proposed method are related to Eqs 4-7? What is the explicit connection between the proposed method and the mentioned interpretability? In particular, the authors should clarify why Eqs. 4-7 can address the challenge of class correction.
>
>
> **R2**: Thank you for the questions regarding Equations 4 to 7. We would like to answer from the following aspects.
>
> * How Eqs. 4-7 are optimized: Eqs. 4-6 are combined as a single multi-objective energy function, Eq. 7, and we rewrote Eq. 7 as described in the experimental setup. The final multi-objective optimization function is: $ min_{\kappa}$  $Z=1-Z^{Acc}+\alpha Z^{COBias} +\beta Z^{Extreme} $, where $\kappa_i$ for class i=1,...,N is chosen from the given set of membership functions, and $\kappa_i = k$ means membership function $f_k$ is chosen. In addition, for a sample, let $p = (p_1, …, p_i, …,p_N)$ be its in-context learned output class probabilities, then these probabilities are transformed by their learned membership functions, according to Eq. 3. The corrected prediction is $\hat{y}= argmax_i$ $f_{\kappa_i} (p_i)$.
>
>
>     The above multi-objective optimization model is solved using the Simulated Annealing (SA) heuristic. The core step of SA is to sample a new solution $\kappa = (\kappa_1, . . . , \kappa_N)$, e.g., (16, 3, …, 8), and evaluate it on the multi-objective function $Z$. If $Z$ is smaller, accept the new solution; otherwise, accept the new solution with an acceptance probability $exp(-\Delta Z/T)$, where T is the temperature at the step.
>
> * Why Eqs. 4-7 can address class corrections: the class corrections made in this paper aims for reducing COBias and improving accuracy. Each equation from 4 to 6 exactly targets one of these two goals. In detail, Eq. 4 targets maximizing overall accuracy, Eq. 5 targets minimizing COBias, and Eq. 6 targets maximizing per-class accuracy, which enforces it to meet a threshold; Eq. 7 combines the three objectives as a multi-objective function.
>
> * The connection between the proposed method and the mentioned interpretability: The membership functions learned by the proposed method enable sample-level interpretability. We highlight the main novelty of this work: the proposed method **enables us to know whether the LLM in-context learns an accurate probability for a class within a given sample**. This is achieved by learning a **correction function** (membership function) for each class, towards the multi-objectives of reducing COBias and enhancing accuracy. For example, if the Don’t Change function is learned for a class, it means the LLM in-context learns an accurate probability for the class; otherwise, a tailored correction is performed by the membership function.
>
> ---
> **Q3**: The performance gain is limited, especially when comparing FuRud to DNIP. For instance, DNIP achieves 6.6% performance gain over BC, while FuRud is 6% more accurate than DNIP. Similar cases can be found in the COBiasmetric.
>
>
> **R3**: Thanks for the comment. Please allow us to correct a point in your comment: FuRud is 9.4% higher than BC and 2.8% higher than DNIP in terms of average accuracy, and “FuRud is 6% more accurate than DNIP” was not a claim by us.
>
> We disagree that FuRud’s performance gain is limited, because **FuRud achieves higher average accuracy gain than DNIP, and comparable average COBias reduction to DNIP**, shown by Table 1 in the paper.
>
> Besides performance gains, we again highlight the main novelty of interpretable debiasing, which enables us to know whether the LLM in-context learns an accurate probability for a class within a given sample. These sample-level interpretations are not available in other methods; for example, DNIP does not explain which class in a sample is accurately learned in context by the LLM.
>
>
> ---
> Thanks again for your constructive comments and your recognition of our efforts. We hope the response can address your concerns.
>
> Best regards,
>
> Authors

---

> > ### Author Response · Authors · 2024-11-25
> > **A kind reminder**
> >
> > Dear Reviewer DNty,
> >
> > Thanks again for your review. We are looking forward to hearing if our response has adequately addressed your concerns.
> >
> > Best wishes,
> >
> > Authors

---

> > > ### Author Response · Authors · 2024-11-30
> > >
> > > Dear Reviewer DNty,
> > >
> > > We sincerely appreciate your valuable contributions as our reviewer and thank you for your time and insights.
> > >
> > > We would like to kindly request a re-evaluation of our work based on the rebuttal and the revised manuscript. We have addressed your concerns with additional texts on the optimization process, and hope these clarifications will resolve any misunderstandings, particularly regarding the explicit connections between the proposed method and interpretability. In a word, the membership functions learned by the proposed method enable sample-level interpretability for tailored corrections.
> > >
> > > Once again, we extend our gratitude for your hard work in making ICLR a success this year.
> > >
> > > Best regards,
> > >
> > > Authors

---

> > > > ### Comment · Reviewer_DNty · 2024-12-03
> > > > **Official Comments**
> > > >
> > > > Hi,
> > > >
> > > > Thanks for your response. It seems that the response to the fourth point is missing.
> > > >
> > > > The motivation of the proposed method is confusing. I cannot figure out why the proposed method works and why methods are designed using such an approach. In particular, the authors should clarify why Eqs. 4-7 can address the challenge of class correction. Since there is space for one page to allow authors to add necessary explanations, authors may consider adding detailed clarifications.

---

> > > > > ### Author Response · Authors · 2024-12-03
> > > > >
> > > > > Dear Reviewer DNty,
> > > > >
> > > > > We sincerely appreciate your reply.
> > > > >
> > > > > >  Motivations of this paper
> > > > >
> > > > > **FuRud is the first method that provides tailored corrections for per-sample, per-class’s probability when tackling the pervasive COBias (class accuracy imbalance) issue in LLMs.** Our approach aims at the mult-objectives of reducing COBias while improving overall accuracy and individual class accuracies, while at the same time, we innovatively leverage fuzzy rules to enable interpretable sample-level corrections.
> > > > >
> > > > > **Strong motivations**: such tailored corrections are challenging, fortunately, fuzzy membership functions exactly help with it. We leverage a set of membership functions to perform corrections. A membership function decides how a class probability in a sample is corrected, which asymmetrically amplifies or reduces different ranges of a class’s probabilities. As a reviewer points out, FuRud addresses both the inter-class surface bias and also the intra-class range-wise influences. We adopt multi-objective optimization to jointly select a set of membership functions for each class towards improving multi-objectives based on COBias and accuracy. Across seven evaluation benchmarks, FuRud achieves state-of-the-art results, while providing the sample-level interpretability.
> > > > >
> > > > > > I cannot figure out why the proposed method works and why methods are designed using such an approach. In particular, the authors should clarify why Eqs. 4-7 can address the challenge of class correction. Since there is space for one page to allow authors to add necessary explanations, authors may consider adding detailed clarifications.
> > > > >
> > > > > It is addressed in **R2**, and we would be happy to discuss more if you have further questions.
> > > > >
> > > > > In addition, we have incorporated additional texts or revisions into the revised manuscript, regarding motivations (line 17, 52-53), why the proposed method works and why methods are designed using such an approach (line 81-104), and the optimization process (line 266-269, 288-298). Thanks again for your kind discussion, and please let us know if more clarifications are needed.
> > > > >
> > > > > Best regards,
> > > > >
> > > > > Authors

---

> ### Author Response · Authors · 2024-12-03
>
> In addition to the above, we have copied and pasted R2 as follows, for your kind reading.
>
> ---
>
> **R2**: Thank you for the questions regarding Equations 4 to 7. We would like to answer from the following aspects.
>
> * How Eqs. 4-7 are optimized: Eqs. 4-6 are combined as a single multi-objective energy function, Eq. 7, and we rewrote Eq. 7 as described in the experimental setup. The final multi-objective optimization function is: $min_{\kappa} Z=1-Z^{Acc}+\alpha Z^{COBias} +\beta Z^{Extreme}$, where $\kappa_i$ for class i=1,...,N is chosen from the given set of membership functions, and $\kappa_i = k$ means membership function $f_k$ is chosen. In addition, for a sample, let $p = (p_1, …, p_i, …,p_N)$ be its in-context learned output class probabilities, then these probabilities are transformed by their learned membership functions, according to Eq. 3. The corrected prediction is $\hat{y}= argmax_i f_{\kappa_i} (p_i)$.
>
>     The above multi-objective optimization function is solved using the Simulated Annealing (SA) heuristic. The core step of SA is to sample a new solution $\kappa = (\kappa_1, . . . , \kappa_N)$, e.g., (16, 3, …, 8), and evaluate it on the multi-objective function Z. If Z is smaller, accept the new solution; otherwise, accept the new solution with an acceptance probability $exp(-\Delta z/T)$, where T is the temperature at the step.
>
> * Why Eqs. 4-7 can address class corrections: the class corrections made in this paper aims for reducing COBias and improving accuracy. Each equation from 4 to 6 exactly targets one of these two goals. In detail, Eq. 4 targets maximizing overall accuracy, Eq. 5 targets minimizing COBias, and Eq. 6 targets maximizing per-class accuracy, which enforces it to meet a threshold; Eq. 7 combines the three objectives as a multi-objective function.
>
> * The connection between the proposed method and the mentioned interpretability: The membership functions learned by the proposed method enable sample-level interpretability. We highlight the main novelty of this work: the proposed method **enables us to know whether the LLM in-context learns an accurate probability for a class within a given sample**. This is achieved by learning a **correction function** (membership function) for each class, towards the multi-objectives of reducing COBias and enhancing accuracy. For example, if the Don’t Change function is learned for a class, it means the LLM in-context learns an accurate probability for the class; otherwise, a tailored correction is performed by the membership function.

---

> > ### Author Response · Authors · 2024-12-04
> >
> > Dear Reviewer DNty,
> >
> > We would appreciate a re-evaluation if you find our rebuttal and the revised manuscript strengthens the paper. We have always appreciated your questions and suggestions.
> >
> > Sincerely,
> >
> > Authors

---

### Official Review · Reviewer_X5d3 · 2024-10-30

**Soundness:** 3
**Presentation:** 1
**Contribution:** 2
**Rating:** 5
**Confidence:** 3

**Summary:**

This paper proposes a debiasing/output-correction method for in-context learning applications, using fuzzy-rule based corrections. It achieves comparable accuracy and debiasing performance to other state-of-the-art (sota) methods on one LLM (Llama-2-13B).

**Strengths:**

- The method is original and has not been investigated prior.
- The range of datasets used is strong, and can provide comprehensive insights (though the range of models used is insufficient).
- The paper is relatively clear despite the specifics being hard to comprehend quickly.
- The topic is relevant and timely.

**Weaknesses:**

1. Only one, relatively small LLM is experimented on (Llama-2-13B). Bias towards a particular class is central to the author's claims (see questions), though this is likely more prevalent in weaker/smaller models, or at least simpler to model/correct for interpretably. The setup is thus fairly simple and it is yet to be seen how biases may present themselves for more sophisticated models, or how it may be fixed.
2. I am unsure about the baselines. They lack proper description based on my reading of the paper. It is hard to understand exactly what they propose. Please see questions.
3. Interpretability is a key motivator but it is not clearly explained why other methods are uninterpretable/why this is a major problem.
4. A major drawback was in places a lack of readability. For instance, it took a long time to try to understand that the process is as follows: full few-shot examples + test questions are passed through the LLM --> probabilities are measured across the answers for each test question --> these are aggregated across all test questions in the multi-objective optimization step (?) --> probabilities are calibrated according to rules learnt. I was confused in Figure 1 about what terms like "optimization set" refer to.

**Questions:**

1. How does a more interpretable baseline of simple calibrated accuracy [1] compare? I understand the new baselines are reported to outperform this, but it is a useful comparison point, and it is not always clear how much trust can be put on previous results which are conducted on presumably different models/datasets (again, it is not reported in the paper here).
2. Why are fuzzy rules any more interpretable than other basic class correction methods? Please provide comparison/examples.
3. Please clarify my understand of the algorithmic process as outlined above in weaknesses.

[1] Calibrate before use: Improving few-shot performance of language models, Zhao et. al., 2021

---

> ### Author Response · Authors · 2024-11-21
> **Response to Reviewer X5d3 (Part 1/2)**
>
> Dear Reviewer X5d3,
>
> Thank you for your precious time and constructive suggestions. We are glad you think our method is original and the topic is relevant and timely.
>
> We would like to answer your concerns in the following.
>
> ---
> **Q1**: Only one, relatively small LLM is experimented on (Llama-2-13B). The range of models used is insufficient. Bias towards a particular class is likely more prevalent in weaker/smaller models, or at least simplerto model/correct for interpretably, and it is yet to be seen how biases may present themselves for more sophisticated models, or how it may be fixed.
>
> **R1**: Thank you for the concern about the broader applications of our work. Indeed, our work is primarily intended for smaller LLMs, due to its broad application prospects and accessibility to many. Our approach, however, can work for any LLM **as long as COBias exists in the model, and output class probabilities are available**. Whether it is relatively small or more sophisticated, it just provides a different starting point for FuRud to optimize for better accuracy and lower COBias.
>
> As for how biases present themselves for larger LLMs, ChatGPT has been known to be prone to the availability bias problem - a tendency to prioritize information that is more easily recalled or readily available in its training data [1]. This bias manifests as over-generations or over-predictions of certain contents, which could also result in imbalanced class accuracies, suggesting that **COBias exists for sophisticated LLMs** like ChatGPT. Note that currently, ChatGPT does not return full probabilities over the entire vocabulary (only top several tokens), making it hard to evaluate.
>
> For more relatively small LLMs, we ran additional experiments on Llama-2-7B and GPT-2-XL. Results are shown in the table below. Accuracy and COBias improvements are seen on these additional models, further showcasing the effectiveness of our approach.
>
> | Prompting Method | Metric | AGNews | DBpedia | SST-5 | TREC | RTE | DDI | PubMedQA | Avg. |
> |---|---|---|---|---|---|---|---|---|---|
> | Llama-2-7B |  |  |  |  |  |  |  |  |  |
> | ICL | Acc | $86.4_{2.5}$ | $88.9_{2.0}$ | $42.1_{11.1}$ | $66.7_{6.6}$ | $66.3_{4.3}$ | $6.7_{0.4}$ | $40.3_{6.7}$ | 56.8 |
> |  | COBias | $14.0_{6.5}$ | $13.5_{2.1}$ | $55.6_{1.5}$ | $33.2_{10.0}$ | $61.6_{10.5}$ | $41.4_{1.7}$ | $40.9_{16.1}$ | 37.2 |
> | FuRud | Acc | $\boldsymbol{88.5_{0.5}}$ | $\boldsymbol{91.5_{0.5}}$ | $\boldsymbol{49.5_{0.7}}$ | $\boldsymbol{73.1_{3.9}}$ | $\boldsymbol{72.7_{1.0}}$ | $\boldsymbol{54.4_{6.4}}$ | $\boldsymbol{55.7_{7.6}}$ | $\boldsymbol{69.3}$ |
> |  | COBias | $\boldsymbol{7.4_{2.5}}$ | $\boldsymbol{8.4_{0.6}}$ | $\boldsymbol{24.0_{1.2}}$ | $\boldsymbol{14.1_{1.9}}$ | $\boldsymbol{4.2_{2.7}}$ | $\boldsymbol{16.9_{5.0}}$ | $\boldsymbol{21.8_{16.6}}$ | $\boldsymbol{13.8}$ |
> | GPT2-XL |  |  |  |  |  |  |  |  |  |
> | ICL | Acc | $52.1_{5.4}$ | $31.8_{9.9}$ | $34.9_{13.7}$ | $27.4_{10.5}$ | $55.4_{1.9}$ | $14.5_{4.4}$ | $55.2_{0.0}$ | 38.8 |
> |  | COBias | $35.5_{11.5}$ | $40.0_{3.6}$ | $48.7_{5.4}$ | $45.6_{8.7}$ | $82.4_{24.5}$ | $40.7_{5.9}$ | $59.4_{12.6}$ | 50.3 |
> | FuRud | Acc | $\boldsymbol{69.0_{0.5}}$ | $\boldsymbol{67.7_{11.8}}$ | $\boldsymbol{43.4_{3.1}}$ | $\boldsymbol{41.7_{2.7}}$ | $\boldsymbol{51.2_{3.7}}$ | $\boldsymbol{53.2_{17.0}}$ | $\boldsymbol{48.4_{0.3}}$ | $\boldsymbol{53.5}$ |
> |  | COBias | $\boldsymbol{7.4_{2.9}}$ | $\boldsymbol{23.0_{6.5}}$ | $\boldsymbol{25.4_{1.4}}$ | $\boldsymbol{30.2_{7.0}}$ | $\boldsymbol{8.9_{3.6}}$ | $\boldsymbol{23.1_{6.5}}$ | $\boldsymbol{17.6_{4.6}}$ | $\boldsymbol{19.4}$ |
>
> Table X5d3-A. FuRud results (in **bold**) on more model sizes and model families. Average score with standard deviation over three runs are reported.
>
> [1] Partha Pratim Ray. ChatGPT: A comprehensive review on background, applications, key challenges, bias, ethics, limitations and future scope. Internet of Things and Cyber-Physical Systems Volume 3 (2023): 121-154

---

> ### Author Response · Authors · 2024-11-21
> **Response to Reviewer X5d3 (Part 2/2)**
>
> **Q2**:  Why are fuzzy rules any more interpretable than other basic class correction methods? How does a more interpretable baseline of simple calibrated accuracy [1] compare? Please provide comparison/examples.
>
> **R2**: Thanks for the thoughtful question. We would like to compare them from the following aspects.
>
> * Calibration methods can hardly provide interpretability for the COBias issue. In this case, methods such as Contextual Calibration [1] are not more interpretable.
>
> * Comparing FuRud to DNIP: both have a debiasing goal on COBias, so the learned weights or rules are wired to provide interpretability on how a class probability should update to mitigate COBias. However, we need to reiterate that, as suggested in Section 5.4, what’s not done in DNIP is the interpretability on why and which classes need corrections, and what sample-specific corrections should be applied, e.g., a tailored correction for per-sample, per-class’s probability.
>
>     * DNIP learns **a per-class correction weight** for **all samples indiscriminately**. What it means is that DNIP treats a class in all samples indifferently, masking the true need of NO correction for some classes in a given sample. For example, for every sample, class A’s probability is multiplied by 0.2 for correction. Such a class’s probability may get updated when it does not have to for some samples.
>
>     * Instead, FuRud learns **a per-class correction function**, i.e., a membership function, which **decides if and how a class’s probability needs correction for each sample**. If correction is needed, the corrected class probability will be **tailored by the membership function**, which can be larger or smaller than its original probability. Therefore, this sample-level interpretation is more interpretable than DNIP, which is a main innovation of this paper.
>
>
> ---
> **Q3**:  I was confused in Figure 1 about what terms like "optimization set" refer to. Please clarify my understand of the algorithmic process as outlined above in weaknesses.
>
> **R3**: Thanks for the question. An optimization set is a full set or a subset (what we call *few-shot optimization*, it is different from 1-shot prompting) of samples used for optimizing the rule selections. Each sample is first prompted in 1-shot manner to obtain output class probabilities. Then, every sample is represented by its class probability vector, and they are input to the multi-objective optimization program, i.e., the multi-objective optimization block illustrated in Figure 1.
>
> The whole process is summarized as follows:
>
> Optimization: 1-shot demonstration + optimization set’s sample’s question is input to the LLM (done for every sample in the optimization set) $\rightarrow$ probabilities are measured across the classes for each optimization set question $\rightarrow$ these are aggregated across all questions in the multi-objective optimization model $\rightarrow$ an optimal membership function is learned for each class
>
> Inference: for each test sample, 1-shot demonstration + test sample’s question is input to the LLM $\rightarrow$ probabilities are measured across the classes for each test set question $\rightarrow$ apply the learned per-class membership function to correct each class’s probability in the sample $\rightarrow$ Get corrected predictions of all test samples and evaluate
>
>
> [1] Tony Z. Zhao, Eric Wallace, Shi Feng, Dan Klein, and Sameer Singh. Calibrate before use: Improving few-shot performance of language models. International Conference on Machine Learning. 2021
>
> ---
> Thanks again for your constructive comments and your recognition of our efforts. We hope the response can address your concerns.
>
> Best regards,
>
> Authors

---

> > ### Author Response · Authors · 2024-11-25
> > **A kind reminder**
> >
> > Dear Reviewer X5d3,
> >
> > Thanks again for your review. We are looking forward to hearing if our response has adequately addressed your concerns.
> >
> > Best wishes,
> >
> > Authors

---

> > > ### Author Response · Authors · 2024-11-30
> > >
> > > Dear Reviewer X5d3,
> > >
> > > We sincerely appreciate your valuable contributions as our reviewer and thank you for your time and insights.
> > >
> > > We have addressed your concerns with additional experimental results and detailed explanations, and hope these clarifications will resolve any misunderstandings, particularly regarding the COBias issue in more LLMs and why fuzzy rules are more interpretable. We kindly request a re-evaluation of our work based on the rebuttal and the revised manuscript.
> > >
> > >
> > > Once again, we extend our gratitude for your hard work in making ICLR a success this year.
> > >
> > > Best regards,
> > >
> > > Authors

---

> > > > ### Author Response · Authors · 2024-12-04
> > > >
> > > > Dear Reviewer X5d3,
> > > >
> > > > We would appreciate a re-evaluation if you find our rebuttal and the revised manuscript strengthens the paper. We have always appreciated your questions and suggestions.
> > > >
> > > > Sincerely,
> > > >
> > > > Authors

---

### Official Review · Reviewer_AN8p · 2024-11-03

**Soundness:** 4
**Presentation:** 4
**Contribution:** 4
**Rating:** 8
**Confidence:** 4

**Summary:**

The paper proposes FuRud, an interpretable fuzzy rule optimization based debiasing method for LLM in-context learning. FuRud addresses both the inter-class surface bias and also the intra-class range-wise influences.

**Strengths:**

I am first of all impressed by the quality of work done, and also the extensiveness of the paper's discussions. The method is also straightforward with promising results and qualitative analyses. Overall, the paper was well written and easy to follow, and many interesting experiments were done.

**Weaknesses:**

- One concern is that the experiments were done on a single model, Llama2-13B. I would like to see if this approach is applicable to other model families and sizes.
- It is well known that the performance of In-Context Learning is largely dependent on how the demonstrative examples are selected. However, I don't think there was any analysis on this, other than on the number of samples used. How were the samples samples selected -- were they selected randomly? Will there be certain example selection strategies that are incompatible with FuRud?

**Questions:**

See Weaknesses

---

> ### Author Response · Authors · 2024-11-21
> **Response to Reviewer AN8p (Part 1/2)**
>
> Dear Reviewer AN8p,
>
> We sincerely appreciate your precious time, insightful questions, and constructive suggestions. We are deeply encouraged by your high recognition of the quality of our work, and also the extensiveness of our paper's discussions.
>
> In the following, we would like to address your concerns.
>
> ---
> **Q1**: Experiments were done on a single model, Llama2-13B. I would like to see if this approach is applicable to other model families and sizes.
>
>
> **R1**: Thank you for the constructive suggestion. We ran experiments of FuRud on two additional models, Llama-2-7B and GPT2-XL.
>
> Results are shown in the table below, demonstrating that FuRud **gains consistent performance improvements on various models**. For example, on Llama-2-7B, FuRud improves accuracy by a relative 22%, and reduces COBias by a relative 63%,  over ICL baselines.
>
> Indeed, our current evaluations are focused on relatively small LLMs, given smaller LLMs’ wide applications and accessibility to many users. However, our approach can also work for larger models, as long as class probabilities are available and the imbalanced per-class accuracy issue exists.
>
> | Model | Metric | AGNews | DBpedia | SST-5 | TREC | RTE | DDI | PubMedQA | Avg. |
> |---|---|---|---|---|---|---|---|---|---|
> | Llama-2-7B |  |  |  |  |  |  |  |  |  |
> | ICL | Acc | $86.4_{2.5}$ | $88.9_{2.0}$ | $42.1_{11.1}$ | $66.7_{6.6}$ | $66.3_{4.3}$ | $6.7_{0.4}$ | $40.3_{6.7}$ | 56.8 |
> |  | COBias | $14.0_{6.5}$ | $13.5_{2.1}$ | $55.6_{1.5}$ | $33.2_{10.0}$ | $61.6_{10.5}$ | $41.4_{1.7}$ | $40.9_{16.1}$ | 37.2 |
> | FuRud | Acc | $\boldsymbol{88.5_{0.5}}$ | $\boldsymbol{91.5_{0.5}}$ | $\boldsymbol{49.5_{0.7}}$ | $\boldsymbol{73.1_{3.9}}$ | $\boldsymbol{72.7_{1.0}}$ | $\boldsymbol{54.4_{6.4}}$ | $\boldsymbol{55.7_{7.6}}$ | $\boldsymbol{69.3}$ |
> |  | COBias | $\boldsymbol{7.4_{2.5}}$ | $\boldsymbol{8.4_{0.6}}$ | $\boldsymbol{24.0_{1.2}}$ | $\boldsymbol{14.1_{1.9}}$ | $\boldsymbol{4.2_{2.7}}$ | $\boldsymbol{16.9_{5.0}}$ | $\boldsymbol{21.8_{16.6}}$ | $\boldsymbol{13.8}$ |
> | GPT2-XL |  |  |  |  |  |  |  |  |  |
> | ICL | Acc | $52.1_{5.4}$ | $31.8_{9.9}$ | $34.9_{13.7}$ | $27.4_{10.5}$ | $55.4_{1.9}$ | $14.5_{4.4}$ | $55.2_{0.0}$ | 38.8 |
> |  | COBias | $35.5_{11.5}$ | $40.0_{3.6}$ | $48.7_{5.4}$ | $45.6_{8.7}$ | $82.4_{24.5}$ | $40.7_{5.9}$ | $59.4_{12.6}$ | 50.3 |
> | FuRud | Acc | $\boldsymbol{69.0_{0.5}}$ | $\boldsymbol{67.7_{11.8}}$ | $\boldsymbol{43.4_{3.1}}$ | $\boldsymbol{41.7_{2.7}}$ | $\boldsymbol{51.2_{3.7}}$ | $\boldsymbol{53.2_{17.0}}$ | $\boldsymbol{48.4_{0.3}}$ | $\boldsymbol{53.5}$ |
> |  | COBias | $\boldsymbol{7.4_{2.9}}$ | $\boldsymbol{23.0_{6.5}}$ | $\boldsymbol{25.4_{1.4}}$ | $\boldsymbol{30.2_{7.0}}$ | $\boldsymbol{8.9_{3.6}}$ | $\boldsymbol{23.1_{6.5}}$ | $\boldsymbol{17.6_{4.6}}$ | $\boldsymbol{19.4}$ |
>
> Table A-AN8p. FuRud results (in **bold**) on more model sizes and model families. Average score with standard deviation over three runs are reported.

---

> > ### Author Response · Authors · 2024-11-21
> > **Response to Reviewer AN8p (Part 2/2)**
> >
> > **Q2**: It is well known that the performance of In-Context Learning is largely dependent on the demonstrative examples. How are the demonstrative examples selected? Will there be certain example selection strategies that are incompatible with FuRud?
> >
> >
> > **R2**: Thank you for the insightful comment. The 1-shot demonstrative example used in this paper was randomly selected from the optimization set. We reported average scores over 3 random seeds to account for variations in demonstrations.
> >
> > To further see how demonstrations in the prompt affect performances, we additionally prompt Llama-2-13B with a more sophisticated demonstration selection strategy: k-shot prompting, where k is the number of classes, and a demonstrative example from each class is represented in the prompt. **FuRud significantly improves accuracy and COBias in this additional setting**, as shown by the table below. To summarize:
> >
> > * Compared to the 1-shot strategy, the k-shot strategy provides a different starting point for FuRud. For example, the average ICL accuracy by k-shot (61.9%) is slightly larger than that obtained by 1-shot (59.4%), and average COBias (25.6%) is smaller than 1-shot (40.5%). FuRud boosts average accuracy to 73.5% and reduces COBias to 13.0%.
> >
> > * **Therefore, different example selection strategies provide different starting points for FuRud to optimize, so no strategies are incompatible.**
> >
> > |  |  | AGNews | DBpedia | SST-5 | TREC | RTE | DDI | PubMedQA | Avg. |
> > |---|---|:---:|---|---|:---:|---|:---:|---|---|
> > | k-shot ICL | Acc | $83.5_{1.5}$ | $95.2_{1.2}$ | $50.3_{2.3}$ | $67.0_{12.7}$ | $75.0_{0.8}$ | $9.7_{1.0}$ | $52.3_{5.3}$ | 61.9 |
> > |  | COBias | $14.9_{5.1}$ | $7.0_{2.2}$ | $36.3_{7.2}$ | $38.2_{5.1}$ | $22.5_{13.2}$ | $39.7_{3.5}$ | $20.9_{4.2}$ | 25.6 |
> > | k-shot FuRud | Acc | $\boldsymbol{88.1_{0.6}}$ | $\boldsymbol{96.6_{0.4}}$ | $\boldsymbol{54.3_{1.3}}$ | $\boldsymbol{77.9_{6.0}}$ | $\boldsymbol{75.9_{4.6}}$ | $\boldsymbol{62.3_{2.1}}$ | $\boldsymbol{59.2_{5.9}}$ | $\boldsymbol{73.5}$ |
> > |  | COBias | $\boldsymbol{7.7_{2.5}}$ | $\boldsymbol{4.4_{0.7}}$ | $\boldsymbol{13.8_{4.1}}$ | $\boldsymbol{11.6_{3.3}}$ | $\boldsymbol{5.0_{1.4}}$ | $\boldsymbol{27.0_{2.2}}$ | $\boldsymbol{21.3_{8.7}}$ | $\boldsymbol{13.0}$ |
> >
> > Table B-AN8p. FuRud results (in **bold**) on k-shot prompting, where k is the number of classes, and a demonstrative example from each class is represented in the prompt. Average score with standard deviation over three runs are reported.
> >
> >
> > ---
> >
> > Thanks again for your constructive comments and your recognition of our efforts. We hope the response can address your concerns.
> >
> > Best regards,
> >
> > Authors

---

> > > ### Comment · Reviewer_AN8p · 2024-11-22
> > >
> > > Thank you for the responses.
> > >
> > > But I think it'd be better to compare FuRud with DNIP, which was the best baseline in the main table.
> > >
> > > Nevertheless, the overall paper quality was good with interesting results. Thus, I lean more towards acceptance.
> > >
> > > I maintain my score.
> > >
> > > Best,

---

> > > > ### Author Response · Authors · 2024-11-23
> > > > **Thank you!**
> > > >
> > > > Dear Reviewer AN8p,
> > > >
> > > > Thank you for your prompt reply and your high recognition of our paper! For a little bit more discussion, we compare DNIP paper’s reported scores on GPT-2-XL and Llama-2-7B with FuRud, and find on both LLMs, FuRud and DNIP’s test performances are similar (the gap between their accuracy, and the gap between their COBias were both less than 1%, with FuRud being slightly better than DNIP on Llama-2-7B), demonstrating the competitiveness of FuRud.
> > > >
> > > > Once again, we sincerely appreciate your time and insightful comments.
> > > >
> > > > Best Regards,
> > > >
> > > > Authors

---

### Official Review · Reviewer_xLmU · 2024-11-04

**Soundness:** 2
**Presentation:** 3
**Contribution:** 2
**Rating:** 5
**Confidence:** 3

**Summary:**

This paper proposes a fuzzy-rule based method to debias probabilities of

**Strengths:**

1. Overall paper is well-written with a novel idea proposed.
2. Experiment results show strong improvements over baselines.

**Weaknesses:**

1. Authors need to clarify the differences between 'debias' and 'calibration' better and earlier (currently mostly discussed in 5.4.). Many references and comparison methods here are 'calibration' based methods such as Batch Calibration ("Batch Calibration: Rethinking Calibration for In-Context Learning and Prompt Engineering.") and "Calibrate Before Use: Im- proving Few-shot Performance of Language Models.".
2. I think this method is not limited to ICL in LLM only. This is not a weakness of the method per se, as if the method can be used somewhere else means it has greater impacts. Does it really not work/applicable to any other classification probabilities setup? My intuition is that with this debias method, it can probably give improvements in other setting as well. But this is the consequence of introducing more computes. Authors need to justify and perform more analysis why this is a method tailored to ICL in LLM to better motivate the paper.
3. When comparing with these calibration method, need to list computation cost, calibration errors as reference metrics to justify the comparison.

**Questions:**

Mostly described in the weakness section

---

> ### Author Response · Authors · 2024-11-21
> **Response to Reviewer xLmU**
>
> Dear Reviewer xLmU,
>
> Thank you for your appreciation and detailed evaluation of our work. We are glad that you think our approach is novel and there are strong improvements over baselines.
>
> We would like to address the concerns, as follows.
>
> ---
> **Q1**: Authors need to clarify the differences between 'debias' and 'calibration' better and earlier (currently mostly discussed in 5.4.)
>
> **R1**: Thanks for the constructive comment. To further explain the terms “debiasing” and “calibration”:
> * Our paper broadly aligns with the **debiasing literature that mitigates different biases in ICL output probabilities**, among which there are **calibration approaches and nonlinear integer programming (NIP) approaches**. The two lines of work significantly differ in their debiasing goals, despite both dealing with biases observed in output classes. Specifically, calibration techniques aim to correct a **model’s imbalanced class prediction**, usually via affine transformations, without considering the influences between classes, whereas NIP techniques further abstract the problem as **imbalanced class accuracy**, and explicitly treat the pairwise class accuracy differences (COBias) as a main debiasing objective.
> * Our approach adopts the mult-objectives based on reducing COBias while improving overall accuracy and individual class accuracies, and innovatively emphasizes on sample-level interpretability leveraging fuzzy rules.
> ---
>
> **Q2**: Does it really not work/applicable to any other classification probabilities setup? Authors need to justify and perform more analysis why this is a method tailored to ICL in LLM to better motivate the paper.
>
>
> **R2**: Thank you for recognizing our work’s potential applications in more settings. We would like to first reiterate our contributions and the novelty of our constructions.
>
> * We have seen that debiasing works have achieved notable performance improvements in correcting LLMs’ ICL outputs. What’s left to be done, and what motivates this work is: **whether the LLM in-context learns an accurate probability for a class within a given sample**. Hence, we are curious to find why and which certain classes need corrections, and more importantly, to have an interpretable transformation for correcting each class. To the best of our knowledge, we are the first to tackle these interpretability challenges.
>
> * We uncover the effectiveness of range-wise probability corrections, leveraging fuzzy membership functions. A membership function is a correction function, and **it directly tells us within a sample, if a class is assigned with an accurate in-context learned probability, highlighting the interpretability**. For example, if the Don’t Change function is selected for a class, it means the LLM in-context learns an accurate probability for the class. When a correction is needed, the function finds the probability range that the class’s probability belongs to, and updates the class’s probability with the returned function value. That is, our approach provides **sample-level interpretations**, which is a main innovation.
>
> * The proposed approach has been evaluated extensively on a diverse range of text classification tasks, which are also compared with state-of-the-art debiasing works.
>
> Back to your concern, ICL in LLMs is probably the most important starting point to showcase our approach, so we dedicate this paper to ICL in LLMs. Moving forward, as you point out, our approach could go beyond LLMs, spurring follow-up works on more modalities.
>
>
>
> ---
> **Q3**: When comparing with the calibration method, need to list computation cost, calibration errors as reference metrics to justify the comparison.
>
>
> **R3**: Thanks for the constructive comment. To address these aspects:
>
> As for computation costs, the computational time of FuRud optimization is **in the scale of minutes**, from several minutes to around 30 minutes, depending on the dataset (e.g., number of classes, optimization set sizes, etc). For the calibration method Batch Calibration (BC), it applies an analytical calculation on all samples’ ICL probabilities, introducing small computational overhead. However, the accuracy improvements and COBias reduction gained by optimization methods can not be replaced by analytical solutions. As for calibration errors, the “calibration” in Batch Calibration means adjusting output probabilities via an analytical calculation, i.e., subtracting the output probability by the mean probability of each class. This calculation does not introduce further bias or calibration error. For other potential calibration errors, we have taken the average over 3 random seeds to account for variations in prompt demonstrative example selection. Mean scores with standard deviation are reported in the paper.
>
> ---
> Thanks again for your constructive comments and your recognition of our efforts. We hope the response can address your concerns.
>
> Best regards,
>
> Authors

---

> > ### Author Response · Authors · 2024-11-25
> > **A kind reminder**
> >
> > Dear Reviewer xLmU,
> >
> > Thanks again for your review. We are looking forward to hearing if our response has adequately addressed your concerns.
> >
> > Best wishes,
> >
> > Authors

---

> > ### Author Response · Authors · 2024-11-30
> >
> > Dear Reviewer xLmU,
> >
> > We sincerely appreciate your valuable contributions as a reviewer and thank you for your time and insights.
> >
> > We have addressed your concerns with additional explanations, and hope these clarifications will resolve any misunderstandings, particularly regarding the motivations. We hope that you might re-evaluate our work based on the rebuttal and the revised manuscript.
> >
> > Once again, we extend our gratitude for your hard work in making ICLR a success this year.
> >
> > Best regards,
> >
> > Authors

---

> > > ### Author Response · Authors · 2024-12-04
> > >
> > > Dear Reviewer xLmU,
> > >
> > > We would appreciate a re-evaluation if you find our rebuttal and the revised manuscript strengthens the paper. We have always appreciated your questions and suggestions.
> > >
> > > Sincerely,
> > >
> > > Authors

---

### Author Response · Authors · 2024-11-22
**Revisions made**

Dear Reviewers and AC,

We truly appreciate your precious time and help in handling this paper. Thank all reviewers for their valuable feedback. We provided detailed answers in the comments below, and addressed the issues in the revised manuscript.

The revised parts in the paper are indicated by blue fonts. We made the following changes:
1. Revised the introduction to better explain the connection between the proposed method and the mentioned interpretability; clarified equations 4 to 7; corrected "per-class accuracy bias" to “the imbalanced class accuracy issue”. (Reviewer DNty)
2. Clarified the whole algorithmic process of FuRud (Reviewer X5d3)
3. Added a discussion about computational costs (Review xLmU)
4. Added discussions and experimental results on more LLMs; using a different demonstrative example selection strategy (Reviewer AN8p)

Best regards,

Authors

---

### Meta-Review · Area_Chair_xzps · 2024-12-17

**Metareview:**

This paper tackles the important issue of bias in ICL for large language models. The authors introduce a novel "fuzzy-rule" based method aimed at reducing class prediction bias while preserving or even improving ICL accuracy. While the experimental results support their claims, the paper has some significant weaknesses.

Strengths:
- The paper addresses a relevant and significant problem in ICL.
- The proposed solution is novel and intuitive.

Weaknesses:
- The paper's clarity and readability need improvement, with reviewers finding the motivations and discussions lacking depth.
- The experiments, while expanded during the rebuttal phase to include more LLMs, still rely on relatively simple models compared to state-of-the-art LLMs. This raises concerns about the generalizability and applicability of the proposed method to more advanced LLMs.
- The performance improvements, particularly in terms of accuracy, are marginal compared with DNIP (acc improvements can be compensated with bias increase).

Despite addressing a worthwhile problem with a novel approach, the paper's weaknesses in clarity, generalizability, and performance gains ultimately lead me to recommend rejection.

**Additional Comments On Reviewer Discussion:**

While the authors have attempted to address concerns about generalizability by providing additional results with more LLMs,  they haven't fully convinced the reviewers that their method extends to more advanced LLMs.  Furthermore,  concerns remain regarding the clarity of the writing and the depth of the discussions. Addressing these issues would significantly enhance the paper's impact and persuasiveness.

---

### Decision · Program_Chairs · 2025-01-22

Reject